# Role of Plasma Angiopoietin-1 and VEGF Levels as Potential Biomarkers in Chronic Central Serous Chorioretinopathy with Macular Neovascularization

**DOI:** 10.3390/ijms251910748

**Published:** 2024-10-06

**Authors:** Michał Chrząszcz, Weronika Pociej-Marciak, Natalia Mackiewicz, Bożena Romanowska-Dixon, Marek Sanak, Sławomir Teper, Maciej Gawęcki, Izabella Karska-Basta

**Affiliations:** 1Clinic of Ophthalmology and Ocular Oncology, University Hospital, 31-501 Krakow, Poland; weronika.pociej-marciak@uj.edu.pl (W.P.-M.); mackiewicz.nat@gmail.com (N.M.); bozena.romanowska-dixon@uj.edu.pl (B.R.-D.); izabella.karska-basta@uj.edu.pl (I.K.-B.); 2Faculty of Medicine, Department of Ophthalmology and Ocular Oncology, Jagiellonian University Medical College, 31-501 Krakow, Poland; 3Molecular Biology and Clinical Genetics Unit, Department of Internal Medicine, Faculty of Medicine, Jagiellonian University Medical College, 31-501 Krakow, Poland; marek.sanak@uj.edu.pl; 4Clinical Department of Ophthalmology, Faculty of Medical Sciences in Zabrze, Medical University of Silesia, 40-055 Katowice, Poland; 5Dobry Wzrok Ophthalmological Clinic, 80-280 Gdansk, Poland; maciej@gawecki.com

**Keywords:** central serous chorioretinopathy, vascular endothelial growth factor, angiopoietin-1, macular neovascularization

## Abstract

To evaluate the plasma levels of angiopoietin-1 and vascular endothelial growth factor (VEGF) and their association with macular neovascularization (MNV) in patients with chronic central serous chorioretinopathy (cCSC). Correlations between plasma cytokine levels, CSC duration, and mean choroidal thickness (CT) were also investigated. Of the 59 patients with cCSC, 10 patients with MNV secondary to cCSC and 10 patients with cCSC without MNV were enrolled in the study. The control group included 15 healthy volunteers matched for age, sex, smoking status, and comorbidities. Chronic CSC was diagnosed based on typical findings on swept-source optical coherence tomography (OCT), fundus fluorescein angiography, and indocyanine green angiography. Additionally, all patients underwent OCT angiography to help detect MNV. Plasma angiopoietin-1 and VEGF levels were assessed using multiplex immunoassay. The plasma angiopoietin-1 levels differed between the 3 groups (*p* = 0.005). The angiopoietin-1 levels were lower in patients with cCSC with MNV than in controls (*p* = 0.006). There were no differences in the plasma VEGF levels between all the 3 groups (*p* = 0.329). The VEGF levels were negatively correlated with mean CT in cCSC patients with MNV (rho = −0.683, *p* = 0.042) but correlated positively with disease duration in patients with cCSC without MNV (rho = 0.886, *p* = 0.003). Our study confirms that MNV is a common complication of cCSC and provides new insights into the role of angiopoietin-1 in cCSC and MNV. Reduced angiopoietin-1 levels in cCSC patients, regardless of MNV status, highlight the importance of the Ang–Tie2 pathway in disease pathogenesis and may point to new therapeutic targets and future novel treatments to improve the management of these patients.

## 1. Introduction

Macular neovascularization (MNV) is one of the most serious vision-threatening complications in chronic central serous chorioretinopathy (cCSC) [1,2]. The diagnosis of MNV in patients with cCSC is often challenging because of overlapping findings on fundus fluorescein angiography (FFA) and indocyanine green angiography (leakage at the level of retinal pigment epithelium [RPE], choroidal hyperpermeability) and a hyperfluorescent pattern both in cCSC with and without MNV [3]. The introduction of optical coherence tomography angiography (OCTA) has significantly increased the sensitivity of MNV detection in patients with CSC. The reported prevalence of MNV confirmed by OCTA ranges from 7.23% to 39.2% in patients with cCSC [4,5]. Nevertheless, the underlying pathogenesis of MNV in cCSC remains unclear. Thus, the optimal protocol for treatment has not been determined yet [6,7].

Spaide [8] and Sacconi et al. [9] postulated that patients with CSC develop MNV due to the dilation of existing vascular channels in the process of vascular endothelial growth factor (VEGF)-independent arteriogenesis, as opposed to VEGF-dependent angiogenesis. It has also been postulated that the dilation of anastomoses between the vortex veins in response to vortex venous congestion results in the formation of dilated vessels—the so-called pachyvessels characteristic of CSC [10]. Until recently, most studies focused on VEGF involvement in vasoproliferative retinal diseases. However, the effect of VEGF is modulated by other angiogenic proteins, particularly angiopoietins [11,12].

Angiopoietin-1 plays a crucial role in the regulation of vascular homeostasis and the pathophysiology of various retinal vascular diseases [11,12,13]. As a key ligand for the Tie2 receptor, angiopoietin-1 is primarily involved in stabilizing blood vessels, promoting endothelial integrity, and reducing inflammatory responses. In the context of retinal diseases such as diabetic retinopathy and age-related macular degeneration, the balance between proangiogenic factors like VEGF and angiopoietins significantly influences disease progression and severity [13,14]. Recent research has highlighted the potential therapeutic benefits of targeting the angiopoietin-1/Tie2 pathway to restore vascular stability and improve outcomes in neovascular diseases. Angiopoietin-1 mediates its vascular stabilizing and anti-inflammatory effects by activating the endothelial Tie2 tyrosine kinase receptor, while angiopoietin-2 is a weak Tie2 agonist with context-dependent activity. By enhancing angiopoietin-1 signaling, it may be possible to mitigate pathological neovascularization and preserve retinal function [13,14,15]. Understanding these dynamics will contribute to the development of more effective treatment strategies and improve management of retinal vascular disorders.

Based on the reported effects of Ang–Tie pathway activation, we hypothesized that it may serve as a new target for the development of novel therapies in this sub-optimally treated disease. Thus, the aim of this study was to assess the presence of MNV in cCSC based on OCTA and to compare the plasma levels of angiopoietin-1 and VEGF in relation to the presence of MNV between patients with cCSC and healthy controls.

## 2. Results

The mean age of the study groups was 54.8 ± 4.8 years for patients with MNV, 54.8 ± 4.7 years for patients without MNV, and 45.3 ± 4.0 for controls. Men constituted 70% of patients in the group with cCSC with MNV, 50% in the group with cCSC without MNV, and 66.7% in the control group. The groups did not differ in age, sex, smoking status, or prevalence of systemic hypertension. Best-corrected visual acuity (BCVA) differed significantly between patients with cCSC and controls (Table 1).

Of the eight women with cCSC, two were postmenopausal, and one was using hormonal contraception. Angiotensin-converting enzyme inhibitors were used by two patients with cCSC with MNV, one patient with cCSC without MNV, and one control; β-blockers, by one patient with cCSC with MNV, one patient with cCSC without MNV and two controls; diuretics, by one patient with cCSC with MNV, two patients with cCSC without MNV, and one control. Finally, sartans were used by one patient with cCSC with MNV and one patient with cCSC without MNV. One patient with cCSC did not use any antihypertensive drugs.

Of the fifty-nine patients with cCSC, ten patients (16.9%) had secondary MNV. Among the ten patients with flat irregular pigment epithelial detachment (FIPED), MNV features on OCTA were revealed in six patients, while four patients with FIPED showed no MNV. None of the controls showed FIPED or MNV. The difference between groups was not significant. Of the eight patients with pigment epithelial detachment (PED), four patients had MNV. None of the controls showed PED. The difference between groups was not significant.

Plasma angiopoietin-1 levels differed between the three groups (*p* = 0.005). They were significantly lower in cCSC patients with MNV (1859.00 pg/mL [251.34–240.00 pg/mL]) vs. controls (3281.00 pg/mL [2597–5069 pg/mL]) (*p* = 0.006). In addition, plasma angiopoietin-1 levels tended to be lower in patients without MNV (2009.00 pg/mL [1146–2637 pg/mL]) vs. controls, but the difference was not significant (*p* = 0.069). Plasma angiopoietin-1 levels were also lower in patients with MNV than in those without, but again the difference was not significant (Figure 1).

Interestingly, there were no differences in plasma VEGF levels between the three groups. The highest median VEGF levels were noted for controls (20.5 pg/mL [3.66–25.15 pg/mL], followed by cCSC patients with MNV (8.06 pg/mL [6.39–13.81 pg/mL]) and those without MNV (6.65 pg/mL [2.83–11.2 pg/mL]) (Figure 2).

There was a negative correlation between the VEGF levels and mean choroidal thickness (CT) in the patients with cCSC with MNV (rho = −0.683, *p* = 0.042). Correlations between the cytokine levels and mean CT are presented in Table 2 and Figure 3.

## 3. Discussion

New imaging modalities such as OCTA have revolutionized our understanding of the pathogenesis and complications of CSC, including MNV. Currently, the interpretation of CSC is largely dependent on advances in evidence-based imaging. The continuous progress in molecular biology has improved our knowledge of CSC and helps support the clinical discovery of new molecular biological markers and therapeutic targets in this disease.

MNV is one of the most common causes of visual acuity deterioration in patients with cCSC [5]. In our study, the lowest BCVA (<0.1) was noted only in cCSC patients with MNV. This is in line with a study by Mrejen et al. [16], which confirmed that MNV is associated with worse visual outcomes in patients with cCSC, as compared with patients without MNV. While the diagnosis of MNV remains challenging, the sensitivity of its detection has increased following the introduction of OCTA [4,17]. The prevalence of MNV confirmed by OCTA ranges from 7.23% to 39.2% of patients with cCSC according to various studies [5,18]. In our study, MNV was detected in 16.9% of patients with cCSC. Savastano et al. [4] reported a higher prevalence of MNV, but their patients were older than those in our study (66.6 ± 10.2 years vs. 54.8 ± 4.8 years). According to Zhou et al. [1], age, recurrence rate, and the presence of leakage on FFA are the main risk factors for MNV in cCSC.

Recent studies on retinal neovascular diseases have primarily focused on the role of VEGF, with anti-VEGF therapies becoming the cornerstone for treating conditions such as age-related macular degeneration (AMD) and diabetic macular edema. However, angiopoietins, particularly angiopoietin-1 and angiopoietin-2, acting through the Tie2 receptor, are increasingly recognized as key regulators of vascular homeostasis, remodeling, and neovascularization. While VEGF promotes pathological neovascularization, angiopoietin-1 stabilizes blood vessels, maintaining endothelial integrity, while angiopoietin-2 destabilizes them, priming the vasculature for VEGF-driven angiogenesis. Thus, the Ang–Tie pathway, together with VEGF, plays an important role in vessel maturation and neovascularization development [19].

To our knowledge, our study is the first to assess plasma levels of these 2 counteracting factors, angiopoietin-1 and VEGF, in cCSC patients with and without MNV in comparison with healthy controls. Angiopoietin-1 levels were shown to be lower in patients with cCSC with and without MNV than in controls. Pairwise comparisons revealed a significant downregulation of angiopoietin-1 in patients with cCSC complicated by MNV compared with controls, while a tendency for downregulation was noted in cCSC patients without MNV vs. controls. The available literature shows that angiopoietin-1 counteracts pathological angiogenesis and results in a quiescent, mature vascular phenotype [12,20]. The Ang–Tie signaling pathway plays a key role in vascular stability, mediated by a balance between the agonistic action of angiopoietin-1 on Tie2 and the antagonistic action of angiopoietin-2 [21]. Preclinical studies suggested that the modulation of the Ang–Tie pathway and inhibition of VEGF can restore vascular stability and reduce pathological neovascularization [22].

Only a few studies assessed angiopoietin-1 and angiopoietin-2 levels in human vitreous, and the results are inconsistent. Regula et al. [23] reported slightly elevated levels of angiopoietin-1 in patients with retinal vein occlusion and reduced angiopoietin-1 levels in patients with proliferative diabetic retinopathy. However, the differences were not significant [23]. Our study adds to the understanding of angiogenic factor imbalances in cCSC, particularly the role of angiopoietin-1. We observed that plasma angiopoietin-1 levels were significantly reduced in cCSC patients with and without MNV compared with controls. This suggests that the downregulation of angiopoietin-1 may be a general feature of cCSC pathology rather than a phenomenon solely associated with the development of MNV. However, the most pronounced reduction was seen in patients with MNV, which may indicate that further depletion of angiopoietin-1 in this subgroup facilitates the development of neovascular complications by promoting vascular instability. We hypothesized that a significant reduction in plasma angiopoietin-1 levels in our patients with cCSC complicated by MNV may result in the loss of the vasoprotective properties of angiopoietin-1 and promote neovascularization. Moreover, cCSC patients without MNV tend to have lower levels of angiopoietin-l than healthy controls, which increases the risk of MNV. Such a rationale is supported by the results of early experimental studies on animal model [24,25]. Subsequent human research revealed the involvement of angiopoietin-1 in retinal and choroidal neovascularization. Nambu et al. [20] reported that elevated angiopoietin-1 expression in human eyes with Bruch’s membrane rupture or retinal ischemia inhibits the development of retinal or choroidal neovascularization.

The broader literature on retinal diseases supports our hypothesis. Emerging treatments that target VEGF and angiopoietin-2 utilize the protective role of angiopoietin-1 by stabilizing the vasculature and preventing VEGF-mediated neovascularization [26,27,28,29,30]. Faricimab is a recently approved dual-specific antibody that inhibits both VEGF-A and angiopoietin-2 [31]. Clinical research supporting this registration indicated potential benefits of inhibiting both these angiogenic factors in the treatment of neovascular AMD. This approach promotes the protective role of angiopoietin-1 in wet AMD [26,27,28]. Our results are consistent with this therapeutic strategy, suggesting that restoring the balance in the Ang–Tie2 pathway may be beneficial in cCSC, particularly in patients at risk for MNV. Importantly, the reduction in angiopoietin-1 levels in cCSC patients without MNV suggests that these individuals may already be in a pro-neovascular state, with further deterioration of the angiogenic balance potentially leading to MNV.

The involvement of VEGF in retinal pathology has been extensively studied due to the widespread use of intravitreal anti-VEGF treatment in retinal and choroidal vascular diseases over the past few decades [18]. In our study, the highest VEGF levels were shown for the control group, but the differences between cases and controls were not significant. This is in line with our previous research and other studies reporting no relationship between MNV and VEGF levels, suggesting that VEGF may not be the primary driver of MNV in cCSC, as opposed to other retinal diseases like AMD [8,32,33]. Mrejen et al. [16] suggested that local anatomical and physiological changes characteristic of CSC, such as choroidal congestion, vasodilation, ischemia, and subsequent changes in the RPE, may lead to elevated levels of VEGF in the sub-RPE compartment. This may explain the link between CSC and the development of MNV [3,34] and also the lack of an association between plasma VEGF levels and MNV in the course of CSC. Thus, our data support the hypothesis that MNV development in cCSC may be linked more closely to mechanical and structural changes in the choroid, compounded by disruptions in the angiopoietin-Tie2 pathway, rather than solely VEGF-driven angiogenesis.

In our study, patients with cCSC complicated by MNV showed the highest plasma VEGF levels and the lowest CT. The inverse correlation between VEGF levels and mean CT in cCSC with MNV further complicates the relationship between VEGF and neovascular development in this disease. While VEGF is known to contribute to vascular remodeling, the lack of a clear association between its levels and MNV development in cCSC suggests that other factors, particularly angiopoietins, may play a more pivotal role. A decrease in CT after anti-VEGF treatment in patients with acute CSC was previously reported [35,36]. Therefore, we hypothesize that plasma VEGF levels may have an indirect effect on the choroid and choriocapillaris. However, further research is needed to provide more data.

Our study has several limitations. First, the small sample size limits the ability to draw definitive conclusions. Second, it was conducted at a single time point and only plasma samples were assessed. Third, we did not measure any additional targets in the angiopoietin pathway, but we acknowledge that such measurements could provide further insights into the reported findings. Fourth, several confounding factors may have affected the levels of angiogenic factors, such as concurrent antihypertensive treatment. Finally, we did not take into account the stage of the menstrual cycle in women of reproductive age during blood sampling, whereas the menstrual cycle is known to affect angiogenic markers. Another limitation is the lack of aqueous humor assessment. Given these limitations, the results should be interpreted as preliminary and warrant further research with a larger sample size and a greater range of tested parameters to validate the findings.

## 4. Materials and Methods

### 4.1. Study Population

In this case–control study, of the 59 patients diagnosed with cCSC an, 10 patients with MNV and 10 patients without MNV fulfilling inclusion and exclusion criteria were included in further analysis. A total of 15 eligible patients were randomly assigned to the control group by computer randomization. Controls were matched for age, sex, smoking status, and comorbidities.

Criteria for inclusion in the study were as follows: age older than 18 years, diagnosis of cCSC, symptoms present for the previous 6 months or longer, and follow-up of 6 months or longer. Exclusion criteria were other eye diseases (e.g., AMG, polypoidal choroidal vasculopathy, glaucoma, or ocular trauma), previous treatments (e.g., anti-VEGF treatment, argon laser, photodynamic therapy, or vitreoretinal surgery), significant media opacity precluding fundus imaging, and use of systemic steroids before the onset of clinical symptoms.

The study was approved by the Jagiellonian University Bioethical Committee (approval no. 122.6120.266.2016), and all patients provided written informed consent to participate in the study.

### 4.2. Clinical Examination

All cases and controls underwent complete ophthalmological examination, including BCVA and intraocular pressure assessment, fundus biomicroscopy, swept-source OCT (Topcon DRI OCT Triton, Tokyo, Japan), and OCTA (Topcon DRI Triton, Japan). Additionally, patients with cCSC underwent FFA and indocyanine green angiography (SPECTRALIS, Heidelberg Engineering, Heidelberg, Germany).

Chronic CSC was diagnosed based on the presence of pachychoroid with dilated outer choroidal vessels associated with persistent serous retinal detachment on OCT lasting more than 6 months; widespread areas of leakage from large areas of RPE damage during the early and late phases on FFA; and areas of persistent dilated choroidal vessels with hyperpermeability (pachyvessels) during the early and middle phases, as well as central hyperfluorescence during the late phase on indocyanine green angiography.

The presence of MNV was confirmed based on OCTA findings including decorrelation of abnormal vascular branching on an en-face slab (defined as the RPE-Bruch’s membrane complex) and/or a decorrelation signal within the FIPED or PED on OCTA B-scans, indicating abnormal vascular flow associated with MNV (Figure 4). The automated layer segmentation was manually reviewed and corrected for the line on the Bruch’s membrane and RPE using Imaginet for Triton version 1.03 (Topcon) software. MNV was visualized using a custom en-face slab, with the upper layer positioned offset at 0 μm from the RPE and the lower layer positioned offset at 15 μm from the Bruch’s membrane. Thus, the slab mainly included the FIPED and allowed us to obtain the best image of MNV. This was preferably conducted on 3 mm × 3 mm OCTA images. However, if the MNV margins extended beyond the boundaries, the 6 mm × 6 mm OCTA scans were used.

Choroidal thickness was assessed using the method previously described by Branchini et al. [16]. Mean CT was defined as the average of measurements from 3 points localized beneath the fovea, as well as 750 μm temporally and 750 μm nasally from the fovea. The measurements were taken by two experienced ophthalmologists (MC and IKB).

### 4.3. Sample Collection

Fasting-blood samples were drawn from the antecubital vein in the morning with minimal stasis into vacutainers containing sodium EDTA anticoagulated (BD Life Sciences, Franklin Lakes, NJ, USA). Plasma was separated by centrifugation for 15 min at 2000× *g*. Magnetic Luminex Performance Assay for human high-sensitivity cytokines (FCSTM09-12; R&D Systems Bio-Techne, Minneapolis, MN, USA) was used to measure VEGF and angiopoietin-1 levels in plasma. This multiplex immunoassay contains premixed fluorogenic beads with monoclonal antibodies against VEGF and angiopoietin-1. Bead-trapped cytokines were detected by biotin-streptavidin sandwich immunocomplex fluorescence. The measurements were taken in plasma diluted 1:2 in the included sample dilution buffer and processed according to the manufacturer’s protocol. Washed beads were counted using a MAGPIX × MAP analyzer (Luminex Corp., Austin, TX, USA), and the mean fluorescence index was used for calculations of cytokine levels from a 7-point standard curves using proprietary software, Milliplex Analyst Version 5.1 (Merck, Darmstadt, Germany).

### 4.4. Statistical Analysis

Qualitative data were presented as counts and percentages. Quantitative data were presented as means and standard deviations for normally distributed variables or medians and interquartile ranges for variables without normal distribution. The normality of quantitative variables was tested using the Kolmogorov–Smirnov test. Intergroup comparisons of qualitative variables were performed using the χ2 test, while the exact χ2 test was used when expected frequencies were lower than 5 in at least 20% of cells. Quantitative variables were compared between groups using the Kruskal–Wallis test. When the comparison of the 3 groups showed a significant *p*-value, a pairwise comparison with Bonferroni correction was performed. The strength of the relationship between quantitative variables was assessed using Spearman’s rho correlation coefficient. A *p*-value of less than 0.05 was considered significant. The study was powered to have an 80% probability of detecting a 15% difference in the mean CT at a *p*-value of 0.05. Statistical analysis was performed using IBM SPSS Statistics 24 for Windows.

## 5. Conclusions

In conclusion, this study confirms MNV as a common complication of cCSC and provides new insights into the role of angiopoietin-1 in cCSC and its potential involvement in MNV development. Reduced angiopoietin-1 levels in cCSC patients, regardless of MNV status, highlight the importance of the Ang–Tie2 pathway in disease pathogenesis. However, the more pronounced reduction in patients with MNV suggests that interventions targeting this pathway, aimed at restoring angiopoietin-1 or inhibiting angiopoietin-2 levels, might help develop novel treatments to improve the management of these patients and prevent vision loss. We hope that the preliminary results of our study will spark further interest into the role of pro- and antiangiogenic factors in the development of CSC and MNV. Future research should further explore this pathway, particularly in larger cohorts and with additional focus on intraocular levels of angiogenic factors.

## Figures and Tables

**Figure 1 ijms-25-10748-f001:**
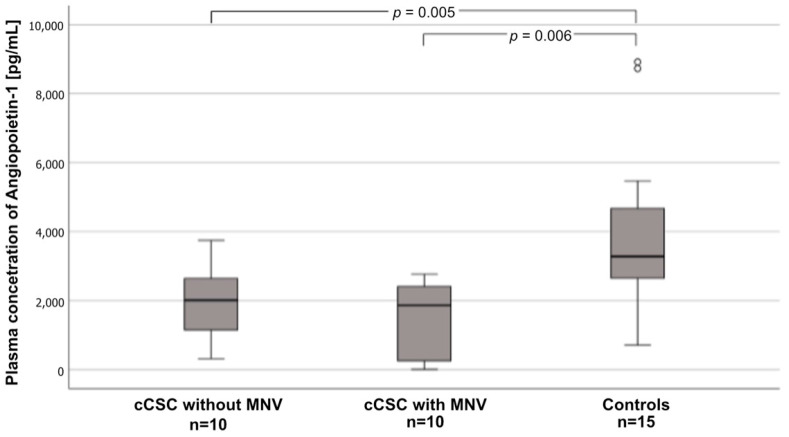
Box-and-whisker plot of plasma angiopoetin-1 levels in patients with chronic central serious chorioretinopathy (cCSC) without macular neovascularization (MNV), patients with cCSC with MNV, and controls. Circles indicate cases distant from an interquartile range of 1.5 to 3.

**Figure 2 ijms-25-10748-f002:**
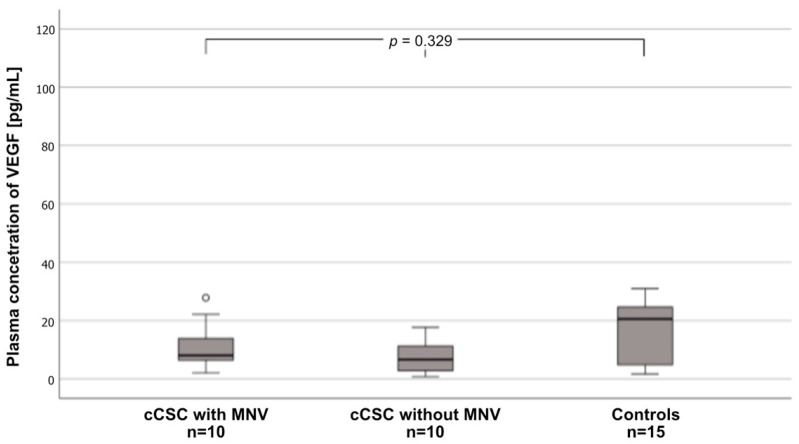
Box-and-whisker plot of plasma vascular endothelial growth factor (VEGF) levels in patients with chronic central serious chorioretinopathy (cCSC) with macular neovascularization (MNV), patients with cCSC without MNV, and controls. Circles indicate cases distant from an interquartile range of 1.5 to 3.

**Figure 3 ijms-25-10748-f003:**
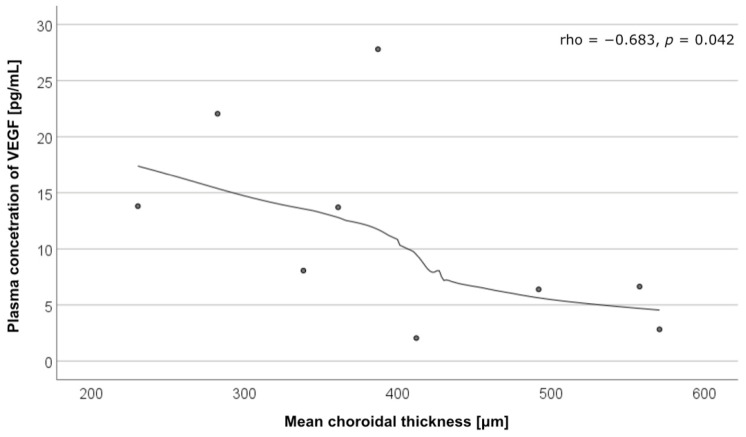
Correlation between plasma vascular endothelial growth factor (VEGF) levels and mean choroidal thickness in patients with chronic central serious chorioretinopathy (cCSC) with macular neovascularization (MNV).

**Figure 4 ijms-25-10748-f004:**
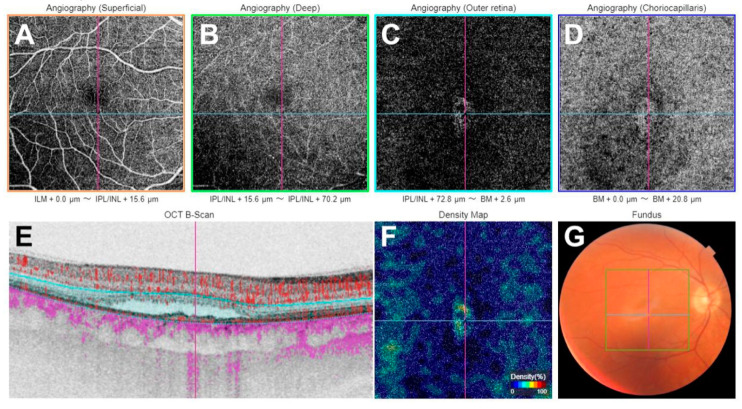
Swept-source optical coherence tomography angiography of macular neovascularization (MNV) secondary to chronic central serous chorioretinopathy: (**A**)—en-face angiogram at the level of the superficial plexus; (**B**)—en-face angiogram at the level of the deep plexus; (**C**)—en-face angiogram at the level of the avascular deep retina with a visible vessel network of MNV; (**D**)—en-face angiogram at the level of choriocapillaris with a visible vessel network of MNV; (**E**)—optical coherence tomography showing flat irregular pigment epithelial detachment (FIPED) and subretinal fluid (SRF) B scan; (**F**)—vessel density map; (**G**)—color fundus picture.

**Table 1 ijms-25-10748-t001:** Demographic and ophthalmological characteristics of patients with chronic serous chorioretinopathy with and without macular neovascularization as well as controls.

Parameter		CSC with MNV (*n* = 10)	CSC without MNV (*n* = 10)	Controls (*n* = 15)	*p* Value
Female sex, n (%)		3 (30.0)	5 (50.0)	5 (33.3)	0.739
Age, y		54.8 ± 4.8	54.8 ± 4.7	45.3 ± 4.0	0.061
Hypertension, n (%)		5 (50.0)	4 (40.0)	4 (26.7)	0.554
Smoking, n (%) (current, former)		1 (10.0)	3 (30.0)	1 (10.0)	0.617
BCVA, n (%)	0.5< and ≤1.0	4 (40.0)	6 (60.0)	15 (100)	0.001
	0.1≤ and ≤0.5	3 (30.0)	4 (40.0)	0 (0)	
	<0.1	3 (30.0)	0 (0.0)	0 (0)	
FIPED		6 (60.0)	4 (40.0)	0 (0)	0.656
PED		4 (40.0)	4 (40.0)	0 (0)	0.542
CT, µm		394.42 [338.67–492.17]	365.33 [329.67–393.33]	313.33 [268.00–364.33]	0.044

Data are expressed as number (percentage) unless otherwise specified. A *p*-value < 0.05 was considered significant. Abbreviations: BCVA, best corrected visual acuity; CT, choroidal thickness; FIPED, flat irregular pigment epithelial detachment; PED, pigment epithelial detachment.

**Table 2 ijms-25-10748-t002:** Correlations between plasma angiopoietin-1 and vascular endothelial growth factor levels and mean choroidal thickness in patients with chronic central serious chorioretinopathy with and without macular neovascularization.

Cytokine	Mean Choroidal Thickness, µm	
	cCSC with MNV (*n* = 10)	cCSC without MNV (*n* = 10)
Angiopoietin-1	rho = −0.261 *p* = 0.467	rho = −0.285 *p* = 0.425
VEGF	rho = −0.683 *p* = 0.042	rho = 0.310 *p* = 0.456

Data are presented as Spearman rho correlation coefficient; *p* < 0.05 was considered significant. Abbreviations: cCSC, chronic central serous chorioretinopathy; MNV, macular neovascular membrane; VEGF, vascular endothelial growth factor.

## Data Availability

Data supporting the findings of this study are available upon request from the corresponding author.

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
