# Peer review of "Role of Plasma Angiopoietin-1 and VEGF Levels as Potential Biomarkers in Chronic Central Serous Chorioretinopathy with Macular Neovascularization"

_ijms, 2024, doi:10.3390/ijms251910748_

Round 1

Reviewer 1 Report

Comments and Suggestions for Authors

The paper titled "Associations Between Macular Neovascularization and Cytokine Levels in Patients with Chronic Central Serous Chorioretinopathy" is aiming to shed the light into the pathogenesis of CSC complicated with MNV. The introduction provides the necessary information for the understanding of the text, the paper is written in a scientifically sound style and the results are clearly presented. However, the authors fail to explain why the plasma angiopoietin-1 level is significantly decreased in both CSC groups, i.e. with and without MNV as compared to controls. As such, this finding does not explain the occurrence of MNV in CSC, but it might offer an explanation for the pathogenesis of CSC in general. Therefore, in its actual form the paper does not address its title, meaning it does not provide an explanation for the association of MNV in patients with CSC.

Author Response

Dear Reviewer,

Thank you for your efforts in evaluating our work and your valuable comments on how to improve it. Below, we present responses to your suggestions and appropriate changes to the manuscript. We hope it will meet with your positive evaluation. 

The paper titled "Associations Between Macular Neovascularization and Cytokine Levels in Patients with Chronic Central SeroDear Reviewer, thank you for your efforts in evaluating our work and your valuable comments on how to improve it. Below, we present responses to your suggestions and appropriate changes to the manuscript. We hope it will meet with your positive evaluation. us Chorioretinopathy" is aiming to shed the light into the pathogenesis of CSC complicated with MNV. The introduction provides the necessary information for the understanding of the text, the paper is written in a scientifically sound style and the results are clearly presented.

Authors: Thank you for acknowledging the importance of this topic. We greatly appreciate your time and feedback. Your detailed comments and suggestions have been invaluable. We have carefully considered and incorporated your suggestions to enhance the quality of the paper. In addition, the paper has been proofread by a medical editor to improve the quality of the English language.

However, the authors fail to explain why the plasma angiopoietin-1 level is significantly decreased in both CSC groups, i.e. with and without MNV as compared to controls.

As such, this finding does not explain the occurrence of MNV in CSC, but it might offer an explanation for the pathogenesis of CSC in general.

Thank you for your feedback. We have revised the discussion as suggested, focusing on the explanation of why plasma angiopoietin-1 levels are significantly reduced in both CSC groups. It must be emphasized that the sample size in our study may limit the ability to draw definitive conclusions. Given this limitation, the results should be interpreted as preliminary and warrant further research with larger sample sizes to validate the findings. We have described this issue as a limitation of our study to ensure that the scope and implications of the study are clearly understood and transparent.

Therefore, in its actual form the paper does not address its title, meaning it does not provide an explanation for the association of MNV in patients with CSC.

Authors: Thank you for your valuable feedback on the title of our study. As suggested, we have revised the title to better reflect the content and scope of our research. The new title is:

“Role of Plasma Angiopoietin-1 and VEGF Levels as Potential Biomarkers in Chronic Central Serous Chorioretinopathy with Macular Neovascularization”

We hope this revised title more accurately reflects the focus of our study. 

Reviewer 2 Report

Comments and Suggestions for Authors

The key role of  angiopoietin-1  in neovascular diseases of the retina has been well established and its targeting has been extensively discussed in light of developing potential therapies in MNV.

In the present study, the authors confirmed the presence of MNV in cCSC patients.

Then they compared the plasma levels of angiopoietin-1 and VEGF in relation to the presence of MNV in a small number of patients with cCSC.

The results of the present work can be summarized as follows:

1.    Plasma ANG-1 levels were lower in cCSC patients with MNV in respect to those without MNV

2.    Plasma levels of VEGF did not differ between the two groups.

Considering that the differences in ANG-1 levels were not so impressive and considering that no differences were determined in VEGF levels then additional targets in the ANG pathway should be measured in order to reach any conclusion in the present work.

This also in light of its first part that was limited to confirm the presence of MNV in cCSC patients.

In this line, the discussion was a sort of revision on previous findings on the role of ANG and VEGF in neovascular diseases of the retina with some omission. The latest findings in the literature of VEGF, angiopoietin and angiopoietin-like factors, and the Ang-2/Tie-2 complex were recently reviewed including an overview of faricimab use in clinical trials.

The authors should better focus the added value of the present work in the broad panorama of the literature on the role of ANG in neovascular diseases of the retina.

Results from clinical trials should be also discussed in light of preclinical findings.

Comments on the Quality of English Language

No comment

Author Response

Dear Reviewer,

Thank you for your efforts in evaluating our work and your valuable comments on how to improve it. Below, we present responses to your suggestions and appropriate changes to the manuscript. We hope it will meet with your positive evaluation. 

The key role of angiopoietin-1 in neovascular diseases of the retina has been well established and its targeting has been extensively discussed in light of developing potential therapies in MNV.

Authors: Thank you for acknowledging the importance of this topic. We greatly appreciate your time and feedback. Your detailed comments and suggestions have been invaluable. We have carefully considered and incorporated your suggestions to enhance the quality of the paper. In addition, the paper has been proofread by a medical editor to improve the quality of the English language.

In the present study, the authors confirmed the presence of MNV in cCSC patients. Then they compared the plasma levels of angiopoietin-1 and VEGF in relation to the presence of MNV in a small number of patients with cCSC.

Thank you for your valuable suggestion. We agree that the sample size in our study is small, which may limit the ability to draw definitive conclusions. Given this limitation, the results should be interpreted as preliminary and warrant further research with larger sample sizes to validate the findings. We have described this issue as a limitation of our study to ensure that the scope and implications of the study are clearly understood and transparent.

The results of the present work can be summarized as follows:

  1. Plasma ANG-1 levels were lower in cCSC patients with MNV in respect to those without MNV
  2. Plasma levels of VEGF did not differ between the two groups.

Thank you for this suggestion. We have revised the results section to address this comment and underline that plasma ANG-1 levels were lower in cCSC patients with MNV vs. those without, but the difference was not significant. If you have any further suggestions on how the results section could be improved, we will be happy to address them.

Considering that the differences in ANG-1 levels were not so impressive and considering that no differences were determined in VEGF levels then additional targets in the ANG pathway should be measured in order to reach any conclusion in the present work.

Thank you for your valuable comment. We acknowledge the importance of measuring additional targets within the ANG pathway to draw more definitive conclusions. Unfortunately, it is not feasible for us to assess these additional targets in the current study, as we do not have access to the necessary data.

This preliminary study sheds some new light on the pathogenesis of cCSC complicated by MNV. However, we plan to conduct further research to build upon our initial findings and provide more insights. We acknowledged the absence of these results in the limitations section of our manuscript.

This also in light of its first part that was limited to confirm the presence of MNV in cCSC patients.

Thank you for your insightful comment. We would like to clarify that the primary focus of our study was to confirm the presence of MNV in patients with cCSC. We recognize that the initial part of our study is limited in scope, and we appreciate your understanding of this focus. If this explanation is not satisfactory, we would be grateful if you could provide further clarification on your question, so that we could further improve our manuscript.

In this line, the discussion was a sort of revision on previous findings on the role of ANG and VEGF in neovascular diseases of the retina with some omission. The latest findings in the literature of VEGF, angiopoietin and angiopoietin-like factors, and the Ang-2/Tie-2 complex were recently reviewed including an overview of faricimab use in clinical trials.

The authors should better focus the added value of the present work in the broad panorama of the literature on the role of ANG in neovascular diseases of the retina.

Results from clinical trials should be also discussed in light of preclinical findings.

Thank you for this valuable comment. We added a few recent publications that are relevant to this topic (references 29-33). Recent studies on retinal neovascular diseases have primarily focused on the role of VEGF, with anti-VEGF therapies becoming the cornerstone for treating conditions like AMD and diabetic macular edema. However, angiopoietins, particularly Ang1 and Ang2, acting through the Tie2 receptor, are increasingly recognized as key regulators of vascular homeostasis, remodeling, and neovascularization. While VEGF promotes pathological neovascularization, Ang1 stabilizes blood vessels, maintaining endothelial integrity, whereas Ang2 destabilizes them, priming the vasculature for VEGF-driven angiogenesis.

Our study adds to the understanding of angiogenic factor imbalances in cCSC, particularly the role of Ang1. We observed that plasma Ang1 levels were significantly reduced in both cCSC groups (with and without MNV) compared to controls. This suggests that the downregulation of Ang1 may be a general feature of cCSC pathology rather than a phenomenon solely associated with the development of MNV. However, the most pronounced reduction was seen in patients with MNV, which may indicate that further depletion of Ang1 in this subgroup facilitates the development of neovascular complications by promoting vascular instability.

This is consistent with the known role of Ang1 in maintaining vascular stability and preventing pathological angiogenesis. Ang1, through its activation of the Tie2 receptor, promotes the maturation and stabilization of blood vessels, counteracting VEGF-driven vascular permeability and neovascularization. Reduced Ang1 levels in both cCSC groups might reflect a vulnerability to vascular leakage and remodeling, but the greater reduction in the MNV group suggests that this imbalance becomes more critical when neovascularization occurs.

The broader literature on retinal diseases also supports this interpretation. Emerging treatments like faricimab, which targets both VEGF and Ang2, capitalize on the protective role of Ang1 by stabilizing the vasculature and preventing VEGF-mediated neovascularization. Our findings align with this therapeutic strategy, indicating that restoring the balance in the Ang-Tie2 pathway could offer benefits in cCSC, particularly for patients at risk of MNV. Importantly, the reduction in Ang1 in cCSC patients without MNV suggests that these individuals may already be in a pro-neovascular state, with further deterioration of angiogenic balance potentially leading to MNV.

Interestingly, we did not observe significant differences in plasma VEGF levels between cCSC patients and controls. This is in line with previous studies suggesting that VEGF may not be the primary driver of MNV in cCSC, as opposed to other retinal diseases like AMD. Instead, our data support the hypothesis that MNV development in cCSC may be linked more closely to mechanical and structural changes in the choroid, compounded by disruptions in the angiopoietin-Tie2 pathway, rather than solely VEGF-driven angiogenesis.

Additionally, the inverse correlation between VEGF levels and mean choroidal thickness in cCSC with MNV further complicates the relationship between VEGF and neovascular development in this disease. While VEGF is known to contribute to vascular remodeling, the lack of a clear association between its levels and MNV development in cCSC suggests that other factors, particularly angiopoietins, may play a more pivotal role.

In conclusion, this study contributes new insights into the role of Ang1 in cCSC and its potential involvement in MNV development. The general reduction of Ang1 in cCSC patients, regardless of MNV status, highlights the importance of this pathway in disease pathogenesis. However, the more pronounced reduction in MNV cases suggests that interventions targeting the Ang-Tie2 axis, aimed at restoring Ang1 levels or inhibiting Ang2, may offer therapeutic promise. Future research should explore this pathway further, particularly in larger cohorts and with additional focus on intraocular levels of angiogenic factors.

This revision integrates the added value of the study within the context of existing literature, demonstrating how it advances the understanding of angiopoietin's role in retinal neovascular diseases and emphasizing the therapeutic implications of these findings. We hope that the preliminary results of our study will spark further interest into the role of pro- and antiangiogenic factors in the development of cCSC and MNV.

Round 2

Reviewer 2 Report

Comments and Suggestions for Authors

The authors must respond to all criticisms

Comments on the Quality of English Language

English requires moderate correction

Author Response

Dear Sir or Madam,

Thank you for the effort you put into reviewing our article. Your comments and suggestions allowed us to improve our work and make it more valuable. Below, we present our responses to your comments - both to your personal comments and to the suggestions in the official table for necessary changes to the manuscript.

We hope that this will allow you to positively evaluate our work. We would be grateful if you could kindly reconsider our manuscript for publication.
Thank you for your time and attention.

Round 3

Reviewer 2 Report

Comments and Suggestions for Authors

No further comments